# Sustainable Financing through Crowdfunding

**Carla Martínez-Climent [1],\*, Ricardo Costa-Climent [2] and Pejvak Oghazi [3]**

1    Faculty of Economics, University of Valencia, 46010 Valencia, Spain
2    Department of Informatics, Linnaeus University, 351 95 Växjö, Sweden; ricardo.costa@lnu.se
3    School of Social Sciences, Sodertorn University, 141 89 Huddinge, Sweden; pejvak.oghazi@sh.se
\*    Correspondence: carla.martinez-climent@uv.es

**Abstract:** The phenomenon of crowdfunding has been widely studied, while the sustainability of crowdfunded ventures is attracting growing interest from academia and society. In light of this interest, we conducted bibliometric analysis to study the relationship between crowdfunding and crowdfunded ventures' sustainability orientation. We analyzed the number of publications, type of publications, and most productive countries, journals, and authors. We also analyzed the most cited articles and examined their approach to sustainability and crowdfunding. The results suggested that a sustainability orientation could bring about change in the current financial and environmental system.

**Keywords:** crowdfunding; sustainability; social; environmental

---

## 1. Introduction

In 1985, Queen, U2, Madonna, Elton John, The Who, Paul McCartney, Bob Dylan, Eric Clapton, and a host of others performed as part of *Live Aid* to help fight poverty and hunger in Africa. Thanks to the powerful mix of performance, technology, and public goodwill, *Live Aid* raised $127 million for famine relief in Africa. More than 30 years have passed since the *Live Aid* concert [1]. Since then, the world has evolved significantly as a result of global technological change. This has affected the world in numerous ways, including the way that companies, individuals, and non-governmental organizations (NGOs) are funded.

Sustainability is a cross-cutting concept with a broad range of implications. Specifically, sustainability relates to social and environmental development [2]. Business practices are important because they affect all involved stakeholders. Some firms promote sustainable innovation in their products, processes, services, and business models. These actions are no less important than the firm's competitiveness and market orientation [3]. Moreover, firms have numerous reasons to promote sustainability. These include economic and ecological motivations [3].

Sustainability affects a range of areas, such as social entrepreneurship, corporate social responsibility [4–8], social innovation [2,9,10], and innovation for sustainable growth [11]. In this paper, we analyzed a specific form of financing (i.e., crowdfunding), which can contribute to sustainable development.

The motivation for this research laid in the need to clarify the nature of the relationship between crowdfunding and the sustainability orientation of crowdfunded projects. To shed light on this relationship, we conducted a literature review based on bibliometric analysis of the linkages the between the terms "crowdfunding" and "sustainability."

## 2. Theoretical Framework

### 2.1. Crowdfunding

Crowdfunding is an innovative form of financing. The protagonists are the members of the crowd, the fundraiser, and the online funding platform that manages flows between the two [4,12]. The main feature of crowdfunding is that it renders traditional financial intermediaries unnecessary. Individuals invest directly in projects to meet the funding needs of entrepreneurs or ventures. In return for making this pledge, backers receive a reward, which may be economic or social [13]. The pledge is made by a relatively small number of backers over the Internet [14–16].

Another feature of crowdfunding that has been highlighted by numerous authors is the interconnection between investors and entrepreneurs on the Internet. These actors contribute in different ways: providing either money or a business idea [17,18]. Accordingly, one of the reasons for the rapid growth of crowdfunding is interaction over the Internet and social networks, as well as the pitching of ventures that takes place through these channels. This has led to the emergence and development of different crowdfunding models. These different types of crowdfunding are based on earlier models, such as microfinancing and cooperatives [19–21]. However, they go beyond these models because this interconnection is used to not only provide financing for ventures and entrepreneurs but also establish relationships with customers and investors, develop products, and test the market. Consumers play a key role because crowdfunding can offer a new communication channel through which firms can generate interest in fledgling products, just as they can identify target customers that demand a given product. In short, crowdfunding offers a tool to create a community, geographically develop networks between backers and creators [22,23], and even generate long-term bonds between consumers, followers, and suppliers [18]. Hence, studies have shown that crowdfunding actually not only removes the need for financial intermediaries but also drives innovation by enabling contact between ventures and consumers [24].

A host of crowdfunding studies have examined the behavior of investors who pledge their money to projects [14], while other studies have focused on the outcome of the post campaign [14,25–28]. Scholars have also studied the specific use of these crowdfunding platforms [29] and even the array of business models that fall under the category of crowdfunding [18] whether these are owned by customers, a third party, or community shares [30].

### 2.1.1. Crowdfunding Models

This section describes the different crowdfunding models. There is a broad spectrum of crowdfunding models. They have diverse features, and their orientation ranges from purely economic to purely social [12]. This typology is clearly determined by the motivations of crowdfunders [31].

Prior to this study, a bibliometric analysis of peer-to-peer lending and equity-based crowdfunding was performed [32]. Peer-to-peer lending, equity-based crowdfunding, reward-based crowdfunding, and donation-based crowdfunding all share common characteristics. For example, all forms of crowdfunding depend on a large number of investors and an online platform to manage interactions between investors and creators [14,26,29,33,34]. Below, we briefly describe each form of crowdfunding.

Peer-to-peer lending is a form of financing that enables loans between individuals without intervention from financial intermediaries. The risk is greater than with other transactions. Accordingly, the return on investment is also higher [34–39].

In equity-based crowdfunding, investors, in exchange for their investment, receive shares in the business project they have pledged to [13,29,33,35,40–42].

When investors receive a token, product, service, or gift in exchange for their pledge to the project, this is known as reward-based crowdfunding [42–46].

Finally, donation-based crowdfunding aims to raise funds to contribute to social causes, such as non-governmental organizations (NGOs). Investors invest in these projects without expecting any economic return. Instead, they seek a social reward by contributing to sustainable development [47–50].

As Figure 1 shows, investors' motivation with each type of crowdfunding was different. In peer-to-peer lending and equity-based crowdfunding, investors were extrinsically motivated, and they hoped to receive an economic reward. In reward-based crowdfunding, investors hoped to receive some sort of material gain, so they were also motivated by extrinsic motivation. In donation-based crowdfunding, however, investors were driven by intrinsic motivation because the reward they hoped to receive was social. Therefore, a priori, it would seem to be more closely related to sustainability than any other form of crowdfunding. However, investors are becoming increasingly motivated by other factors, such as philanthropy [51].

| Crowdfunding based on financial return | **Peer-to-peer lending** (or lending-based crowdfunding) |
| | Extrinsic motivation through monetary reward |
| | **Equity-based crowdfunding** |
| | Extrinsic motivation through economic reward |
| Crowdfunding based on non-financial return | **Reward-based crowdfunding** |
| | Extrinsic motivation through material gain |
| | **Donation-based crowdfunding** (or patronage) |
| | Intrinsic motivation of investors (social return) |

**Figure 1.** Spectrum of crowdfunding models. Adapted from Lam and Law [29].

### 2.1.2. Crowdfunding and ICT

Numerous studies have focused on the effects of information and communication technologies (ICTs) on crowdfunding. For example, Kromidha and Robson [52] affirmed that fundraisers and backers who identified with their projects within their own social networks achieved higher rates of backers or pledges. Zheng et al. [53] suggested that social network relationships of entrepreneurs in terms of their obligations to fund other entrepreneurs, as well as the project's shared meaning between the funders and fundraisers, had crucial effects on online reward-based crowdfunding performance in both the U.S.A. and China.

Mollick [14] reported that the amount raised through crowdfunding was strongly influenced by the entrepreneur's number of friends on social networks. From another perspective, Bechter et al. [54] reported that two well-known platforms (Facebook and Twitter) were important for entrepreneurs who aimed to link with friends and fans who were interested in providing information and financial support.

Zheng et al. [53] categorized social networks into two types with respect to crowdfunding. The first refered to the social network platform where the entrepreneur presented the project (e.g., Kickstarter), whereas the second referred to the entrepreneur's embeddedness in other third-party social networks (e.g., Twitter and Facebook). In both categories, ICTs, social networks, and the online community played vital roles in strengthening the entrepreneur's social capital [55].

### 3. Methodology

We conducted a bibliometric analysis of publications in the Web of Science (WoS). The goal was to review the literature on the linkages between crowdfunding and sustainability. The WoS database enables identification of scientific publications indexed in high-impact journals that have undergone a publication process designed to ensure the high standards of the research and the content contained therein [56].

The aim of this paper was to gain a better understanding of the linkages between sustainability and crowdfunding. Therefore, we performed a study based on the keywords of "crowdfunding" and

"sustainability" or "crowdfunding" and "sustainable." We also included the term "crowd-funding" in the search to avoid introducing bias.

To achieve our research aims, we performed a systematic literature review based on bibliometric analysis. Such analysis consists of analyzing publications on a specific theme using a database that enables the measurement of citations and published documents to interpret advances in the field and the degree of academic interest these might have [57–60]. We, therefore, analyzed the metadata that related to the names of journals, authors, countries, type of document, and area of knowledge, and we observed the most relevant phenomena. We adopted the WoS terminology, with the term "article" specifically denoting journal articles published in WoS journals. Proceedings papers are explicitly referred to as such. This process provided insight into future lines of research [61–63].

## 4. Results

This section presents the results of our analysis of WoS data on the relationship between "crowdfunding" and "sustainability."

Figure 2 shows that the number of published documents has grown since 2013. As shown by the previous figure, the phenomenon of crowdfunding has been increasingly linked to sustainability. The concern for sustainability is reflected in Figure 2. The number of crowdfunding publications has been on an upward trend, as reported in previous research [32]. The number of publications on crowdfunding and sustainability has also been increasing. The heightened attention of researchers studying or analyzing this topic has managerial implications. These are discussed in the conclusions section.

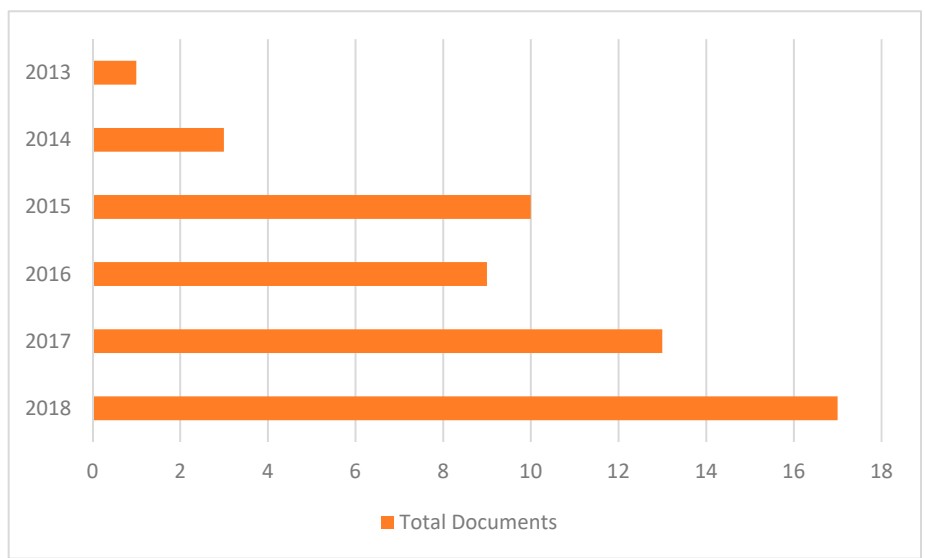

**Figure 2.** Publications by year.

As Table 1 shows, more than 69% of the publications on this topic were scientific articles. Furthermore, 12 of the 53 documents were proceedings papers. Only four reviews were published in journals indexed in the WoS.

**Table 1.** Type of document.

| Type of Document | Publication |
|:---:|:---:|
| Article | 37 |
| Proceedings paper | 12 |
| Review | 4 |
| Total | 53 |

Proceedings papers were published at the conferences shown in Figure 3:

The 9ᵗʰ International Forum on Knowledge Asset Dynamics (IFKAD),

The 3ʳᵈ International Symposium in Computational Economics and Finance,

The ACM SIGCHI Conference on Human Factors in Computing Systems (CHI),

The 17ᵗʰ International Academic MindTrek Conference on Making Sense of Converging Media,

The 32ⁿᵈ International Conference on Education and Research in Computer Aided Architectural Design in Europe (eCAADe),

The International Conference on Modern Management, Education Technology, and Social Science (MMETSS),

The 11ᵗʰ European Conference on Innovation and Entrepreneurship (ECIE),

The 5ᵗʰ International Conference Innovation Management, Entrepreneurship and Sustainability (IMES),

The 8ᵗʰ International Conference on Advances in Information Technology (IAIT),

The Portland International Conference on Management of Engineering and Technology (PICMET),

The SocInfo International Workshops and 20ᵗʰ International Conference on Engineering Design (ICED).

**Figure 3.** Conferences that have published proceedings papers on crowdfunding and sustainability.

The WoS categories in which proceedings papers have been published vary considerably. These categories are Business, Finance > Economics, Engineering, Industrial > Engineering, Electrical & Electronic > Operations Research & Management Science, Computer Science > Cybernetics > Information Systems> Computer Science > Theory & Methods > Architecture. This shows that crowdfunding has been a cross-cutting topic and that research on crowdfunding has been of interest to scholars from numerous knowledge areas.

Interestingly, in the publications classified as reviews, the areas studied were more diverse than those mentioned earlier. More specifically, the most cited review, which had 24 citations and was written by Lam and Law in 2017 [31], was indexed in the category of Green & Sustainable Science & Technology and Energy & Fuels. The other reviews were indexed in the categories of Biotechnology & Applied Microbiology; Genetics & Heredity; Green & Sustainable Science & Technology; Energy & Fuels; and Environmental Sciences. Thus, these publications definitely appeared to be related to sustainability, technology, and energy development.

The bulk of the publications were articles. Of the 53 analyzed documents, 37 were articles. In turn, of these 37 articles, eight were published in WoS categories of Business and Environmental Sciences. The category with the next highest number of articles was Green Sustainable Technology and Management with six publications. The categories of Education and Educational Research, Engineering Environmental, Environmental Studies, and Information Science and Library Science were also among the areas where three articles have been published. Therefore, crowdfunding, as well as its relationship with sustainability, has been studied in these areas. Later in this paper, the articles and the fundamental features of the most relevant articles are analyzed.

Table 2 shows the countries with the highest productivity. The country with most publications was China, with seven documents and 24 citations. The country with the most citations was Germany, with 31. The country with the most citations per document was Australia, with 7.67.

Figures 4 and 5 display the differences between the countries with the most publications (U.S.A. and Canada) and the countries with the most citations of these publications (Germany, Canada, and China).

**Table 2.** Countries with the highest productivity.

| Rank | Country | TP | TC | C/P | h index |
|------|---------|----|----|-----|---------|
| 1 | China | 7 | 24 | 3.43 | 1 |
| 2 | USA | 7 | 13 | 1.86 | 2 |
| 3 | England | 5 | 10 | 2 | 1 |
| 4 | Germany | 5 | 31 | 6.2 | 2 |
| 5 | Italy | 5 | 1 | 0.2 | 1 |
| 6 | Australia | 3 | 23 | 7.67 | 1 |
| 7 | Belgium | 3 | 1 | 0.33 | 1 |
| 8 | Canada | 2 | 26 | 13 | 2 |
| 9 | Spain | 2 | 6 | 3 | 2 |
| 10 | India | 2 | 6 | 3 | 2 |

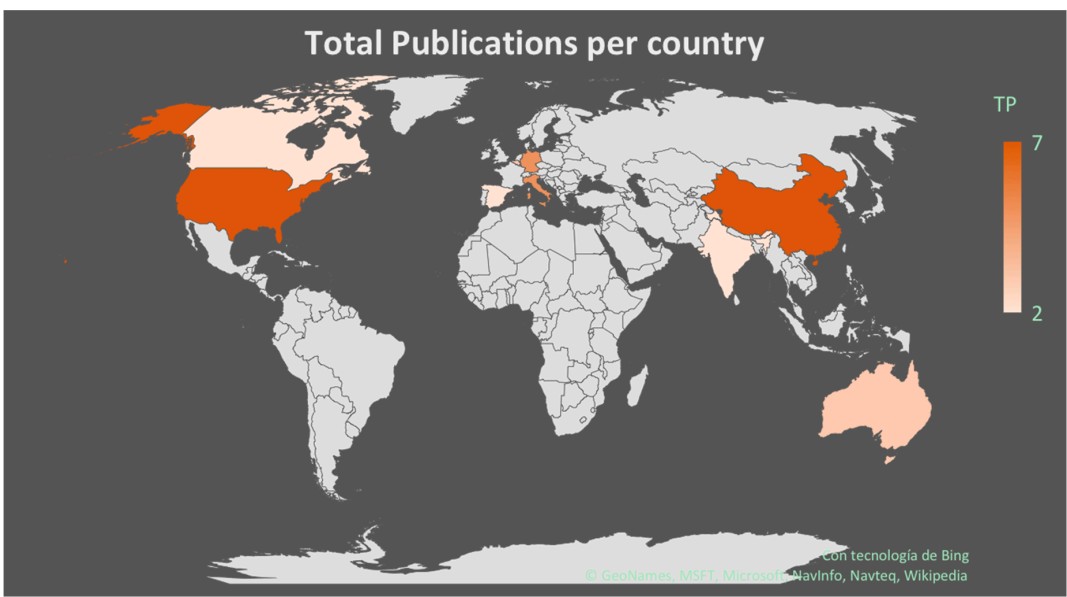

**Figure 4.** Publications by country.

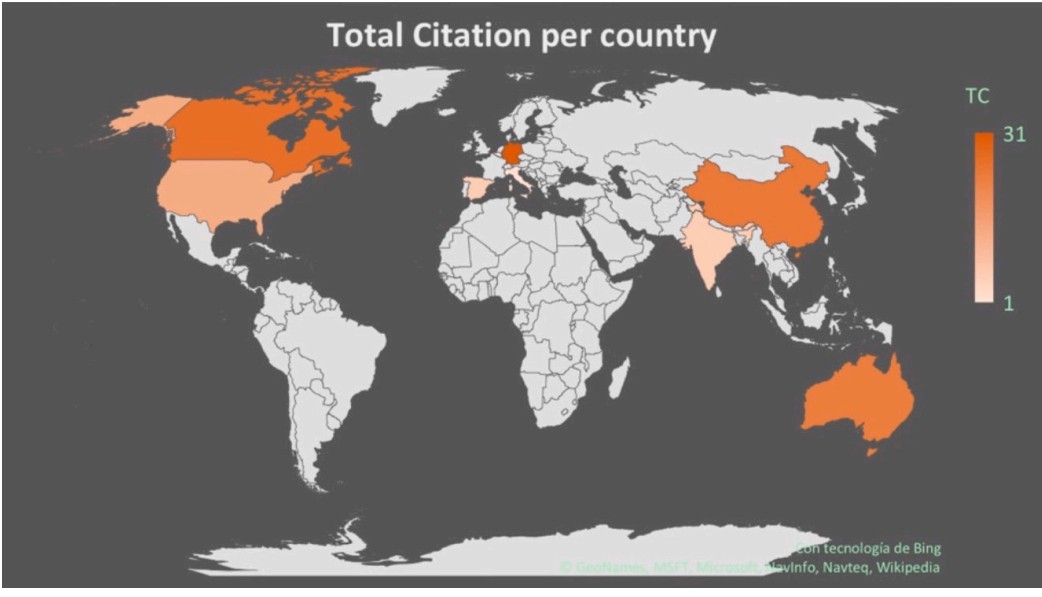

**Figure 5.** Total citations by country.

The most productive journals on crowdfunding and sustainability are shown in Figure 6. We selected the journals with the most cited articles on crowdfunding and sustainability. As explained earlier, the number of publications on this topic was incipient because it has been growing, although there were relatively few publications. However, despite having published relatively few documents (three), the *Journal of Cleaner Production* received 43 citations. *Renewable Sustainable Energy Reviews* also published just three documents, which received 23 citations. These two journals are included in the first quartile of the Journal Citation Reports (JCR) and are indexed in the WoS categories of Engineering, Environmental Sciences, Green & Sustainable Science & Technology, Green & Sustainable Science & Technology, and Energy & Fuels.

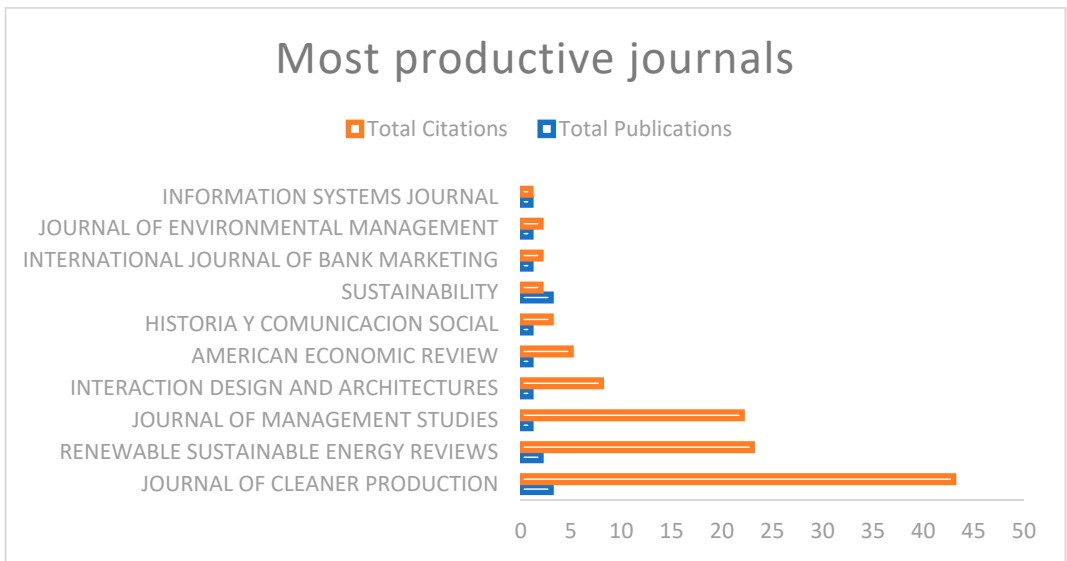

**Figure 6.** Most productive journals.

Notably, the journals *Interaction Design and Architectures* and *Historia y Comunicación Social*, which belong to the emerging index of the WoS, also published works on crowdfunding and sustainability. These journals are indexed in the Emerging Sources Citation Index (Education, Educational Research, Economics) and the Arts and Humanities Citation Index (Film, Radio, Television, History), receiving eight and three citations, respectively.

Table 3 shows the authors with the most citations of publications on crowdfunding and sustainability. No author published more than two documents on crowdfunding and sustainability, and authors with high h indices and total citations leaned toward the study of crowdfunding and sustainability. This finding was important because it showed the interest of researchers with a broad experience in this incipient topic. Hörisch had the most citations (23) in this area, with two publications on the topic of crowdfunding and sustainability. Calic had most citations per document, with 22 citations for a single document. Authors with high h indices, such as Domingo-Ferrer, who had an h index of 27 and 2931 total citations, published one document on crowdfunding and sustainability. The h index was previously used in research as a measure of productivity and impact in the academic community [61,62]. TP refers to the Total Publications, TC refers to Total Citation, CF&S refers to Crowdfunding & Sustainability.

**Table 3.** Most productive authors.

| Rank | Author | Country | TP-CF&S | TC-CF&S | C/P-CF&S | H INDEX-CF&S | H | TP | TC |
|------|--------|---------|---------|---------|----------|--------------|---|----|----|
| 1 | Hörisch, J. | Germany | 2 | 23 | 11.5 | 1 | 7 | 19 | 177 |
| 2 | Calic, G. | Canada | 1 | 22 | 22 | 1 | 2 | 4 | 32 |
| 3 | Light, A. | England | 2 | 9 | 4.5 | 1 | 7 | 31 | 208 |
| 4 | Domingo-Ferrer, J. | Spain | 1 | 3 | 3 | 1 | 27 | 222 | 2931 |
| 5 | Pak, B. | Belgium | 2 | 1 | 0.5 | 1 | 1 | 19 | 4 |
| 6 | Chen. J. | China | 1 | 1 | 1 | 1 | 19 | 92 | 1276 |
| 7 | Benlian, A. | Germany | 1 | 1 | 1 | 1 | 12 | 39 | 637 |
| 8 | Chen, J. | China | 1 | 1 | 1 | 1 | 13 | 84 | 409 |
| 9 | Coutts, C. | U.S.A. | 1 | 1 | 1 | 1 | 13 | 33 | 389 |
| 10 | Bojei, J. | Malaysia | 1 | 1 | 1 | 1 | 2 | 8 | 38 |

We analyzed the 10 most cited articles on crowdfunding and sustainability (Table 4). Four of these articles [2,24,51,64] focused on the Kickstarter platform. As noted by Calic, Goran, Mosakowski, and Elaine [2], Kickstarter is unique because it does not allow philanthropic donations. This policy goes against the preconceived idea that some may have of sustainability. As we have already explained, there are different types of crowdfunding. By establishing this policy, Kickstarter ensures that its business model is not based on donations by specializing in reward-based crowdfunding.

**Table 4.** Most cited paper.

| Rank | PY | TC | AF | SO | Summary | Platforms | Conclusions |
|------|----|----|----|----|---------|-----------|-------------|
| 1 | 2015 | 23 | Hörisch | Journal of Cleaner Production | The paper explores the relationships between environmental orientation of crowdfunding projects and funding success. The paper answers how environmental orientation of crowdfunding projects influences their likelihood of successfully receiving funding. | Indiegogo | Environmental orientation of CF projects currently cannot be observed to be positively related to the success of CF projects. Projects in categories that generate a tangible outcome (e.g., books and videos) are more likely to achieve their funding goals. Non-profit projects tend to be more successful. |
| 2 | 2016 | 22 | Calic, Mosakowski | Journal of Management Studies | The authors study whether and how a sustainability orientation affects entrepreneurs' ability to acquire financial resources through crowdfunding. | Kickstarter | (1) A sustainability orientation positively affects funding success of CF projects, and (2) this relationship is mediated by project creativity and third-party endorsements. A sustainability orientation may matter for creativity within new ventures. |
| 3 | 2016 | 19 | Vasileiadou, Huijben, Raven | Journal of Cleaner Production | What evidence is there that crowdfunding for renewable energy projects has stabilized as a niche and has the potential to break through the energy and financial regimes? | All online crowdfunding platforms in Netherlands* | Evidence of crowdfunding for renewable electricity niches is reported, but the scale remains low. There is limited indication of stabilization of learning processes. With respect to heterogeneity in funders' motivations, normative and gain considerations prevail. Moreover, reward or donation models seem to attract a primarily green crowd. |
| 4 | 2015 | 10 | Jian, Shin | Mass Communication and Society | The authors seek to identify the key motivations behind readers' donations to a pioneering crowdfunded journalism website: Spot.Us. | Spot.Us | Belief in freedom of content, altruism, and contributing to the community were the strongest self-reported motivations by donors of crowdfunded journalism. However, fun and supporting family and friends emerged as clear predictors of high levels of contributions. |
| 5 | 2015 | 8 | Light, Miskelly | Interaction Design and Architectures | The authors explore the idea of sharing culture. They examine the approach of one digital service, regarding sharing as both environmentally and socially sustaining. The paper examines definitions of sharing and explores the positioning of a crowdfunding service. | Patchwork Present | The authors argue that there is a huge, hybridized space, which includes networked services that are disintermediated, thus allowing for new peer-to-peer provision. However, there is no sharing economy, and a belief in one is potentially detrimental to community activity. |

**Table 4.** *Cont.*

| Rank | PY | TC | AF | SO | Summary | Platforms | Conclusions |
|---|---|---|---|---|---|---|---|
| 6 | 2016 | 6 | Gleasure, Feller | Journal of the Association for Information Systems | The authors theorize how anchor values evolve. They analyze how a group of backers on Kickstarter initially embraced the Oculus Rift project, how the relationship changed over time, and how and why these backers responded on hearing news of the sale of Oculus VR to Facebook. | Kickstarter: The Oculus Rift project | The first major implication is to demonstrate the strong role of organizational identity on the crowdfunding process as an input and an output. The paper shows how to move beyond one-to-one dyadic interpersonal relationships and allows researchers to explore hidden inter-group factors that may enhance or limit the use of crowdfunding technologies. |
| 7 | 2017 | 5 | Strausz | American Economic Review | The authors characterize efficient outcomes in the presence of entrepreneurial moral hazard, consumers' private information about demand, and entrepreneurs' private information about cost structure. | Kickstarter | Crowdfunding in the presence of moral hazard and private cost information is unable to attain efficiency in general. It can be thought of as a complement rather than a substitute for traditional venture capital. |
| 8 | 2017 | 3 | Nigussie, Domingo-Ferrer, Sanchez, Osmani | Review of Managerial Science | The authors analyze the investment crowdfunding industry and propose solutions that can neutralize the fear and mistrust effects underlying its market to make it strictly co-utile. | Kickstarter | The market inefficiency arising from fear and mistrust effects, in addition to asymmetric information, limits the applicability of crowd-based financing. |
| 9 | 2015 | 2 | Marakkath, Attuel–Mendes | International Journal of Bank Marketing | This paper discusses how the regulatory environment can be a fundamental constraint or lever in defining the scope of operations of social innovation. | Kiva | There is a need for the specific legal status of crowdfunding platform social ventures, meeting their need to protect their social image while attracting funds. Relaxing the regulations could lead to an expansion of certain types of crowdfunding, particularly those aimed at entrepreneurship, such as equity-based crowdfunding. |
| 10 | 2018 | 2 | Moon, Hwang | Sustainability | The aim of the paper is to identify the factors that influence backers of technology projects through crowdfunding platforms, analyze connections, and establish the usefulness of crowdfunding as a viable funding alternative. | No concrete platform of reward-based crowdfunding | Social influence, effort expectancy, and perceived trust significantly affect the use intention of backers of crowdfunded appropriate technology projects. |

The most cited articles were in the categories of Green & Sustainable Science & Technology; Engineering, Environmental, Environmental Sciences, Business; Management, Communication, Education & Educational Research; Computer Science, Information Systems; Information Science & Library Science; Economics, Film, Radio, Television and History.

*Sustainable Crowdfunding*

We studied the relationship between sustainability and crowdfunding by examining the 53 documents yielded by the bibliometric search.

The study of the relationship between entrepreneurship and sustainable development was the basis for analyzing sustainability. Similarly, to analyze entrepreneurship and its relationship with sustainability and the environment, the specific context must be considered [21]. Hörisch [13] studied the influence of the environmental orientation of crowdfunding projects on campaign success. Hörisch [15] concluded that this influence could not be generalized because the study only showed a positive relationship between the success of crowdfunding campaigns and proposals that generate tangible products. Calic and Mosakowski [2] interpreted the sustainability orientation of crowdfunding projects as a combination of environmental and social considerations. These projects should benefit and protect the environment while improving the lives of people.

Vasileiadou, Huijben, and Raven [18] depicted renewable energy crowdfunding as a new business model. The authors affirmed that this sustainability orientation of crowdfunding projects could change the established financial and energy system.

Jian and Shin [65] studied the website Spot.Us, a donation platform devoted to support journalism. They reported a relationship with sustainability orientation because they defined journalism as a collective good (i.e., goods that can be enjoyed by everybody, such as clean air or a shared knowledge system, like Wikipedia). They identified the factors that encouraged donations, concluding that neither altruism nor freedom of expression is a decisive factor when deciding whether to make donations. Instead, having fun and supporting family and friends were clear predictors of high levels of contributions.

Light and Miskelly [66] defined sustainability as the idea of sharing as an alternative to private property. By sharing, it is possible to split costs and allocate resources in a different way, giving rise to a hybrid space where the concepts of environmental, social, and economic well-being could be implemented.

Nigussie, Domingo-Ferrer, and Sanchez, Osmani [51] analyzed the factors that elicited satisfaction and fear in investors because information asymmetries created inefficiencies in the crowdfunding market. They linked sustainability to crowdfunding by focusing on the crowdfunding business model from the viewpoint of co-utility: "Co-utility is a new concept in which the best way of serving one's own interest is to help in one or more other peers' interest fulfillment" [51] (p. 418).

Marakkath and Attuel-Mendes [67] analyzed the effect of a regulatory environment on operations that sought social innovation. The sustainability focus of the article was based on the idea that social agents should pursue an economic and social mission. The authors concluded that there is a need to create legislation that regulates social operations that maintain the social image, while attracting funding to fulfil the firm's mission. They proposed the relaxing of regulations to protect certain types of crowdfunding, particularly equity-based crowdfunding.

Dilger, Jovanovic, and Voigt [12] studied the range of energy cooperative business models and the role of crowdfunding to improve the problems raised by these business models. They developed the concept of sustainable economics because, in the context energy, there are certain relevant factors to study, such as the source of energy, technical solutions, and energy consumption by businesses. They concluded that cognitive barriers are negative aspects of applying crowdfunding. However, the cooperatives that they studied verified that crowdfunding could play a fundamental role to overcome the challenges that energy cooperatives face.

Moon and Hwang [23] identified a series of factors that influence appropriate technology investors. Appropriate technology aims to bring about social innovation, contributing to developing a local and cultural environment. By performing an analysis of the links between factors that contribute to appropriate technology, the authors established that crowdfunding is a useful tool to finance sustainable projects. They also proposed that reward-based crowdfunding is regularly employed to obtain financing for projects that are less viable in the current system, such as non-profit or artistic projects, whose end goal is not to provide a non-economic return [23].

Walthoff–Borm, Vanacker, and Collewaert [28] studied the result of projects financed using equity-based crowdfunding in the areas of financial performance and innovation performance. In this paper, sustainability is interpreted as the preservation of equity-based crowdfunding projects through investor projection to avoid adverse selection problems.

## 5. Conclusions

In this paper, we analyzed 53 documents that explored the relationship between crowdfunding and sustainability. In one form or another, these documents examined the effect of a sustainability orientation on different crowdfunded projects. The first conclusion was that the definition of sustainability covered a range of areas, with some authors considering economic sustainability. We went further, considering manuscripts that focused on sustainability from a social and environmental perspective to address the established system leading us to climate change, the depletion of the planet's natural resources, and the preservation of the social differences that exist in society.

Accordingly, there is a latent need to seek different forms of organization and execution. The search for solutions from a sustainable approach could encourage outside-the-box thinking that contributes to productivity, social innovation, and highly creative solutions [2].

The bibliometric analysis showed that the year with most publications on the topic of crowdfunding and sustainability was 2018, with 17 publications. The most published type of document over the years was the research article, with 37 documents. China had the highest research productivity, with seven publications that have received 24 citations. The country with the most citations was Germany, with 31. The country with the most citations per document was Australia, with 7.67.

The scientific journal with the most publications was *Sustainability*. This was followed by the *Journal of Cleaner Production*, which, despite having published just three documents, received 43 citations. *Renewable Sustainable Energy Reviews* also published three documents, which received 23 citations. The *Journal of Cleaner Production* and *Renewable Sustainable Energy Reviews* were positioned in the top quartile of the JCR and were indexed in the WoS categories of Engineering, Environmental Sciences; Green & Sustainable Science & Technology; Green & Sustainable Science & Technology; Energy & Fuels.

With respect to the most prolific authors, Hörisch, who published two papers on crowdfunding and sustainability, had the most citations (23) in this area. Calic received the most citations per document of any authors, with 22 citations for a single document.

Finally, the crowdfunding and sustainability articles that received most citations appear in Table 4. Four of these articles [2,24,51,64] focus on the Kickstarter platform. The fact that Kickstarter does not allow donations with philanthropic ends implies that sustainability in crowdfunding operations need not be linked to philanthropy or donations. Rather, sustainability is a cross-cutting concept that should form part of the full range of crowdfunding operations and models.

With respect to the sustainability orientation studied in the 53 documents, some authors affirmed that crowdfunding can reshape the financial and energy system [12,18]. Others claimed that crowdfunding contributes to enabling everybody to enjoy collective goods [66], such as journalism [65], because costs are shared and social, economic, and environmental well-being are promoted. Sustainability orientation was related to social innovation through appropriate technology [23], even with co-utility [51]. Furthermore, several articles [28,68] studied the effect of regulations on crowdfunding and social innovation, which is also a type of sustainability.

However, not all the results revealed a positive relationship between a sustainability orientation and crowdfunding campaign success. For example, Hörisch [15] found that such campaigns must generate physical products to be successful.

*Live Aid* [1] was just an example of the fight against inequality and the efforts to contribute to sustainable development. Thirty years on, new initiatives such as crowdfunding with a sustainability orientation have similar objectives. Consumers, investors, firms, the government, and others can reshape the reality of climate change and social inequality by taking responsible, sensible actions and decisions.

*Managerial Implications and Future Research*

In this paper, we examined the approaches to sustainability and crowdfunding. One key idea of crowdfunding is the bypassing of banks in the financial system to obtain funds for entrepreneurs, firms, and individuals seeking capital.

Banks are increasingly incorporating practices related to corporate social responsibility to cope with calls from society for banks to contribute to sustainable development [67]. However, new forms, such as crowdfunding, are prevailing, and sustainable practices financed by crowdfunding that bypass the established system are being embraced [69].

Thus, the establishment of crowdfunding as part of the system can lead to bypassing the banks and the incorporation of sustainability concerns in the form of commitment to the environment and society, which will promote the distribution of capital [70]. Today, we are witnessing change. The concentration of capital is increasing in multinational companies, which is leading to greater differentiation between social classes and the concentration of wealth. Nevertheless, crowdfunding can contribute to sustainability. It is necessary to establish controls to minimize the risks borne by investors and entrepreneurs [71]. Projects will thereby be more likely to succeed, and the needs of both parties will be met.

Future research should seek evidence of the real contribution of crowdfunding to sustainability in environmental, as well as social, terms.

**Author Contributions:** Conceptualization has been done by C.M.-C., R.C.-C. and P.O.; Methodology, R.C.-C. and P.O.; Investigation, C.M.-C.; Resources, P.O.; Writing—original draft preparation, C.M.-C.; Writing—review and editing, C.M.-C. and R.C.-C.; Visualization, R.C.-C.; Supervision, P.O.; Project administration, P.O.

**Acknowledgments:** The authors gratefully acknowledge financial support from the Chair "Entrepreneurship: Being student to entrepreneur"—Grupo Maicerías Españolas-Arroz DACSA, Cátedra de la Universitat de València "Excelencia y Desarrollo en Emprendimiento: de Estudiante a Empresario"—Grupo Maicerías Españolas-Arroz DACSA.

**Conflicts of Interest:** The authors declare no conflict of interest.

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
