# Peer review of "Sustainable Financing through Crowdfunding"

_sustainability, doi:10.3390/su11030934_

Round 1

Reviewer 1 Report

The paper is interesting as it shed light on the relationship between crowdfunding and the sustainability orientation of crowdfunded projects. Nonetheless, some issues should be addressed before publication:

1. Introduction

The Introduction clearly synthesize the purpose of the paper. The gap is clearly exposed.

There are different types of CF. Are you focusing on financial-return CF (P2P, ECF…)?

Is there any previous bibliometric research on CF in general? You should cite previous research focusing on this as it increases the rigor of your claims.

2. Theoretical Framework

The theoretical framework is well developed. However, I would ask for the inclusion of an epigraph with regard to the implication of CF with ICT (Information and Communication Technology).

3. Methodology

You could improve the description of the methodology.

4. Results

One of the most interesting parts of the paper is the 4.1. Sustainable Crowdfunding. However, you could also create a cluster in order to find relationships between the different papers analyzed on the paper.

5. Conclusions

Authors could have included a section named “Managerial Implications and Future Research”, as it allow the audience to understand the implications from the findings of the research.

6. Minor issues to be considered

·         Lines 97 ff.: For "equity-based crowdfunding", you should consider the following source:

Niemand, T., Angerer, M., Thies, F., Kraus, S., & Hebenstreit, R. (2018). Equity crowdfunding across borders: a conjoint experiment. International Journal of Entrepreneurial Behavior & Research, 24(4), 911-932.

·         Lines 110 ff.: For "reward-based crowdfunding", you should consider the following source:

Kraus, S., Richter, C., Brem, A., Cheng, C. F., & Chang, M. L. (2016). Strategies for reward-based crowdfunding campaigns. Journal of Innovation & Knowledge, 1(1), 13-23.

·         Lines 130 ff: For the methodology of a bibliometric citation analysis, please quote:

Ferreira, J. J., Fernandes, C. I., & Kraus, S. (2017). Entrepreneurship research: Mapping intellectual structures and research trends. Review of Managerial Science, 1-25, https://doi.org/10.1007/s11846-017-0242-3  

·         Lines 142 ff., Figure 2: I would suggest deleting the phrase "Publication Years" from the figure, as there is already a headline on top of it.

·         Line 150 ff. (and before): I have a problem with the word "Article". I suppose what you mean is a JOURNAL ARTICLE, since also proceedings papers are ARTICLES. I would therefore recommend renaming this into "Journal Article".

·         General comment: You also interchangably use "paper" or "article" when talking about YOUR (/this) Work. I would also recommend using only 1 term for this.

7. Native speaker-proofreading:

I would also recommend a native-speaker language check/proofreading before submitting the revised version.

Author Response

Reviewer 1

1. Introduction

The Introduction clearly synthesize the purpose of the paper. The gap is

clearly exposed.

There are different types of CF. Are you focusing on financial return

CF (P2P, ECF…)?

Answer: We thank the referee for the valuable contribution.

Regarding the question: we are not focusing in the financial return CF, we searched for CF in general and filter by the sustainability of the actions.

Is there any previous bibliometric research on CF in general? You should cite

previous research focusing on this as it increases the rigor of your claims.

Answer: On page 3 it is statedPrior to this study, a bibliometric analysis of peer-to-peer lending and equity-based crowdfunding was performed [30].”

2. Theoretical Framework

The theoretical framework is well developed. However, I would ask for the

inclusion of an epigraph with regard to the implication of CF with ICT

(Information and Communication Technology).

Answer: We agree. An epigraph has been included in the third page in order to clarify the implication of CF and ICT.

3. Methodology

You could improve the description of the methodology.

4. Results

One of the most interesting parts of the paper is the 4.1. Sustainable

Crowdfunding. However, you could also create a cluster in order to find

relationships between the different papers analyzed on the paper.

Answer: Thank you, we agree that the 4.1. is one of the most relevant parts of our paper.

5. Conclusions

Authors could have included a section named “Managerial Implications and

Future Research”, as it allow the audience to understand the implications

from the findings of the research.

Answer: We thank you for your detailed comments and suggestions. We have included this sections.

6. Minor issues to be considered

· Lines 35 ff.: I would advise also considering the following source

from the target journal here:

Kraus, S., Burtscher, J., Vallaster, C., & Angerer, M. (2018). Sustainable

Entrepreneurship Orientation: A Reflection on StatusQuo

Research on

Factors Facilitating Responsible Managerial Practices. Sustainability,

10(2), 444.

· Lines 97 ff.: For "equitybased

crowdfunding", you should consider

the following source:

Niemand, T., Angerer, M., Thies, F., Kraus, S., & Hebenstreit, R. (2018).

Equity crowdfunding across borders: a conjoint experiment. International

Journal of Entrepreneurial Behavior & Research, 24(4), 911932.

· Lines 110 ff.: For "rewardbased

crowdfunding", you should consider

the following source:

Kraus, S., Richter, C., Brem, A., Cheng, C. F., & Chang, M. L. (2016).

Strategies for rewardbased

crowdfunding campaigns. Journal of

Innovation & Knowledge, 1(1), 1323.

· Lines 130 ff: For the methodology of a bibliometric citation analysis,

please quote:

Ferreira, J. J., Fernandes, C. I., & Kraus, S. (2017). Entrepreneurship

research: Mapping intellectual structures and research trends. Review of

Managerial Science, 125,

https://doi.org/10.1007/s1184601702423.

Kraus, S., Filser, M., O’Dwyer, M., & Shaw, E. (2014). Social

entrepreneurship: an exploratory citation analysis. Review of Managerial

Science, 8(2), 275292.

Answer: We have included the proposed references as them add value to the research.

· Lines 142 ff., Figure 2: I would suggest deleting the phrase

"Publication Years" from the figure, as there is already a headline on top

12/1/2019 MDPI | Reply review report

https://susy.mdpi.com/user/manuscripts/review/7046397?report=3648576 3/3

of it.

Thank you, it has been deleted.

· Line 150 ff. (and before): I have a problem with the word "Article". I

suppose what you mean is a JOURNAL ARTICLE, since also

proceedings papers are ARTICLES. I would therefore recommend

renaming this into "Journal Article".

Answer: We appreciate your observation regarding the use of the word "article" in the manuscript. You are quite right that it could be interpreted as more broadly referring to journal articles as well as proceedings articles. While we are inclined to defend our use of this term purely to ensure consistency with the terminology employed by the WoS, we feel it would be helpful to define the term "article" in the manuscript as follows: "We have adopted the WoS terminology, with the term "article" specifically denoting journal articles published in WoS journals. Proceedings papers are specifically referred to as such." We hope this solution will be to the reviewer's satisfaction.

· General comment: You also interchangably use "paper" or "article"

when talking about YOUR (/this) Work. I would also recommend using

only 1 term for this.

Answer: We have chosen only one term as proposed by the referee.

7. Native speakerproofreading:

I would also recommend a nativespeaker

language check/proofreading

before submitting the revised version.

Answer: Thank you, it has been done.

Reviewer 2 Report

Thank you for the opportunity to review your manuscript, "Sustainable Financing through Crowdfunding."  This was a very interesting paper, and it should provide researchers with the opportunity to quickly understand the current state of the Crowdfunding literature as it regards to sustainable financing.  I do have a few comments concerning the paper that should help to strengthen your position and contribution.  I hope these prove useful.

To begin, your paper is very interesting, but your contribution, I believe, can be improved.  You end the abstract with the sentence, "The results...financial and environmental system."  I don't believe that your results actually suggest this.  I do believe that the results of the analysis are important, but you should consider an answer to the question, "so what do these results tell us concerning future research and usefulness of crowdfunding, and where should we go from here."  To say that a "sustainability orientation can bring about change in the current financial and environmental system" just does not seem to fit with where your paper was heading.  Additionally, your conclusion paragraph is very nearly another recap of what the results have stated, yet I would like to know where this indicates we should go in the future with research and direction.  Perhaps thinking through the implications of this study can allow you to add a paragraph or two at the end of the conclusions concerning the implications as well as a better constructed sentence at the end of the abstract.

Additionally, I would like to see more development of the final paragraph prior to the Methodology section.  You are really developing justification for a study like this here, and you should give this more attention and development.  

Why do you state the first sentence under Results that states "Authors should discuss...of pervious". I believe you need to delete this.

The paragraph on page 4 beginning with line 146 "Figure 1 shows that the number..."  needs some explanation.  It is not apparent why the "phenomenon of crowdfunding is increasing linked to sustainability" just from looking at the figure.  Much more explanation is necessary.

On page 5, you should take the information from paragraph from lines 158-176 that talk about the conferences and categories and put this into a chart or graph.  This is a long paragraph that could be organized in a quick and easy to read chart or graph.

Also, I am not sure of the significance of the paragraphs about Countries in lines 204-211.  You need to give some background or explain the importance of this information as well as offer the insight that you believe this brings to the paper.

On line 242, you need to reword or explain what "...are leaning toward the study of crowdfunding and sustainability" means.

I also believe that section 4.1 could probably be included in Table 4 by including another column that offers your analysis of the paper.  To have an entire section devoted to this is not effective, and it would be nice to see this quickly in relationship to the overview that you have already given in the table.  This will make a large table, but it should be beneficial.

There are a few cites that I think you should consider also including in your paper.  Please incorporate these in your manuscript.

Motylska-Kuzma, A. (2018). Crowdfunding and Sustainable Development. Sustainability10(12), 4650.

Bi, S., Liu, Z., & Usman, K. (2017). The influence of online information on investing decisions of reward-based crowdfunding. Journal of Business Research71, 10-18.

Angerer, M., Niemand, T., Kraus, S., & Thies, F. (2018). Risk-reducing options in crowdinvesting: An experimental study.Journal of Small Business Strategy, 28(3), 1-17.

Thank you again for allowing me to review your manuscript.  I do hope that my suggestions prove useful as you develop the paper.

Author Response

Reviewer 2.

Thank you for the opportunity to review your manuscript, "Sustainable

Financing through Crowdfunding." This was a very interesting paper, and it

should provide researchers with the opportunity to quickly understand the

current state of the Crowdfunding literature as it regards to sustainable

financing. I do have a few comments concerning the paper that should help

to strengthen your position and contribution. I hope these prove useful.

To begin, your paper is very interesting, but your contribution, I believe, can

be improved. You end the abstract with the sentence, "The results...financial

and environmental system." I don't believe that your results actually suggest

this. I do believe that the results of the analysis are important, but you

should consider an answer to the question, "so what do these results tell us

concerning future research and usefulness of crowdfunding, and where

should we go from here." To say that a "sustainability orientation can bring

about change in the current financial and environmental system" just does

not seem to fit with where your paper was heading. Additionally, your

conclusion paragraph is very nearly another recap of what the results have

stated, yet I would like to know where this indicates we should go in the

future with research and direction. Perhaps thinking through the implications

of this study can allow you to add a paragraph or two at the end of the

conclusions concerning the implications as well as a better constructed

sentence at the end of the abstract.

Additionally, I would like to see more development of the final paragraph prior

to the Methodology section. You are really developing justification for a

study like this here, and you should give this more attention and

development.

Why do you state the first sentence under Results that states "Authors

should discuss...of pervious". I believe you need to delete this.

Answer: We thank you for your detailed comments and suggestions. It has been done, thank you for noticing this mistake.

The paragraph on page 4 beginning with line 146 "Figure 1 shows that the

number..." needs some explanation. It is not apparent why the

"phenomenon of crowdfunding is increasing linked to sustainability" just from

looking at the figure. Much more explanation is necessary.

Answer:

On page 5, you should take the information from paragraph from lines 158176

that talk about the conferences and categories and put this into a chart

or graph. This is a long paragraph that could be organized in a quick and

easy to read chart or graph.

Answer: Thank you, it has been changed into a graph

Also, I am not sure of the significance of the paragraphs about Countries in

lines 204211. You need to give some background or explain the importance

of this information as well as offer the insight that you believe this brings to

the paper.

Answer: We have used the terminology employed by the WoS. Country refers to the affiliation of researchers who conducted studies on CF & Sustainability.

On line 242, you need to reword or explain what "...are leaning toward the

study of crowdfunding and sustainability" means.

Answer: We intend to explain that renowned authors are interested in the topic and they are deciding to research about CF & Sustainability.

I also believe that section 4.1 could probably be included in Table 4 by

including another column that offers your analysis of the paper. To have an

entire section devoted to this is not effective, and it would be nice to see this

quickly in relationship to the overview that you have already given in the

table. This will make a large table, but it should be beneficial.

Answer: We thank the referee for the proposed comments. However, the other reviewer stated that “this part was one of the most interesting parts of the paper”. So we have decided to maintain it.

There are a few cites that I think you should consider also including in your

paper. Please incorporate these in your manuscript.

MotylskaKuzma,

A. (2018). Crowdfunding and Sustainable

Development. Sustainability, 10(12), 4650.

Bi, S., Liu, Z., & Usman, K. (2017). The influence of online information on

investing decisions of rewardbased

crowdfunding. Journal of Business

Research, 71, 1018.

Angerer, M., Niemand, T., Kraus, S., & Thies, F. (2018). Riskreducing

options in crowdinvesting: An experimental study.Journal of Small Business

Strategy, 28(3), 117.

Thank you again for allowing me to review your manuscript. I do hope that

my suggestions prove useful as you develop the paper.

Answer: We thank you for your detailed comments and suggestions. In light of your comments, we have made several changes, which in our opinion have notably improved the manuscript.